# Pharmacological and In Silico Analysis of Oat Avenanthramides as EGFR Inhibitors: Effects on EGF-Induced Lung Cancer Cell Growth and Migration

**DOI:** 10.3390/ijms23158534

**Published:** 2022-08-01

**Authors:** Lorenza Trabalzini, Jasmine Ercoli, Alfonso Trezza, Irene Schiavo, Giulia Macrì, Andrea Moglia, Ottavia Spiga, Federica Finetti

**Affiliations:** 1Department of Biotechnology, Chemistry and Pharmacy, University of Siena, 53100 Siena, Italy; jercoli92@gmail.com (J.E.); alfonso.trezza2@unisi.it (A.T.); schiavoirene@gmail.com (I.S.); giulia.macri@student.unisi.it (G.M.); ottavia.spiga@unisi.it (O.S.); 2Department of Agriculture, Forest and Food Sciences, Plant Genetics and Breeding, University of Torino, 10095 Torino, Italy; andrea.moglia@unito.it

**Keywords:** avenanthramides, EGFR, lung cancer, avenanthramide C, docking simulation, classical molecular dynamics simulation, steered molecular dynamics simulation

## Abstract

*Avena sativa* L. is a wholegrain cereal and an important edible crop. Oats possesses high nutritional and health promoting values and contains high levels of bioactive compounds, including a group of phenolic amides, named avenanthramides (Avns), exerting antioxidant, anti-inflammatory, and anticancer activities. Epidermal growth factor receptor (EGFR) represents one of the most known oncogenes and it is frequently up-regulated or mutated in human cancers. The oncogenic effects of EGFR include enhanced cell growth, angiogenesis, and metastasis, and down-regulation or inhibition of EGFR signaling has therapeutic benefit. Front-line EGFR tyrosine kinase inhibitor therapy is the standard therapy for patients with EGFR-mutated lung cancer. However, the clinical effects of EGFR inhibition may be lost after a few months of treatment due to the onset of resistance. Here, we showed the anticancer activity of Avns, focusing on EGFR activation and signaling pathway. Lung cancer cellular models have been used to evaluate the activity of Avns on tumor growth, migration, EMT, and anoikis induced by EGF. In addition, docking and molecular dynamics simulations showed that the Avns bind with high affinity to a region in the vicinity of αC-helix and the DGF motif of EGFR, jeopardizing the target biological function. Altogether, our results reveal a new pharmacological activity of Avns as EGFR tyrosine kinase inhibitors.

## 1. Introduction

Oats (*Avena sativa* L.) are a whole grain cereal containing high levels of bioactive compounds including phenolic acids, tocopherols, and alk(en)ylresorcinol derivatives, and are a unique source of avenanthramides (Avns), a group of phenolic amides which are not present in other cereals. Structurally, Avns are composed by an anthranilic acid linked through an amide bond to a hydroxycinnamic acid. Oats contain a unique group of approximately 40 different types of Avns that differ on the substitution patterns, the most abundant being AvnA (N-(4′-hydroxycinnamoyl)-5-hydroxyanthranilic acid), Avn-B (N-(4′-hydroxy-3′-methoxycinnamoyl)-5-hydroxyanthranilic acid), and AvnC (N-(3′-4′-dihydroxycinnamoyl)-5-hydroxyanthranilic acid) [1,2].

Pharmacological studies have shown that Avns possess pleiotropic bioactivities, including antioxidant, anti-inflammatory, anti-proliferative, anti-fibrotic, anti-itching and anti-atherogenic properties, with consequent major beneficial health effects [3,4,5,6]. Several indications suggest that Avns may elicit anticancer activity by targeting and modulating different molecular signaling [1,7,8]. In the last few years it has been described that Avns inhibit cancer cell proliferation [9,10,11], epithelial mesenchymal transition (EMT) [7] and regulate apoptosis by affecting DDX3 [12], sirtuin 1 [13], MAPK/NF-kB [14], and miR-19-3p/Pirh2/p53 [15] pathways.

Epidermal growth factor receptor (EGFR) is frequently involved in the pathogenesis and progression of tumor and represents an important pharmacological target in cancer therapy [16,17]. The oncogenic effects linked to EGFR activation or overexpression include increased cell growth, invasion and metastasis, induction of the angiogenic process and cell de-differentiation. Several pharmacological approaches have been developed and, at the present, both small tyrosine kinase inhibitors (TKIs) (gefitinib, erlotinib, Osimertinib and dacomitinib) and monoclonal antibody (cetuximab) are used in clinical practice [18]. However, the benefits linked to EGFR inhibition may be lost after few months of patient treatment due to the onset of chemoresistance [19,20].

The high number of studies on potential drugs obtained from natural sources have showed that natural molecules may represent a productive reservoir of novel drugs due to their chemical characteristics and biological activity. At present, most of the synthetic EGFR TKIs are identified as ATP-competitive inhibitors of EGFR. Therefore, it is of great relevance to identify potential EGFR-TKIs from natural compounds with more effective activity and reduction of side effects on normal tissues.

This study aims to investigate the antitumor effects of AvnA and C on lung cancer cell models treated with EGF and the molecular mechanisms underlying the antitumor activity. Our results demonstrate that AvnA and C inhibit EGF-induced cell growth and migration by inhibition of EGF signaling pathways.

In addition, in vitro and in silico studies suggest that Avns can strongly block the EGF-induced EGFR phosphorylation by their potential interaction in a sensing region of EGFR, located close to αC-helix and the DGF motif, further proving the ability of Avns to inhibit EGFR. 

Taken together, our data highlight that Avns are very promising bioactive compounds that can be further developed and exploited in cancer therapy as EGFR inhibitors.

## 2. Results

### 2.1. Avenanthramides Inhibit EGFR-Induced Lung Cancer Cell Growth 

In a previous study we investigated the anticancer activity of Avns in colon cancer models [7]. In this work we further explored the antitumor potential of Avns by analyzing their effects on lung cancer cells and in EGF driven signaling, representing the engine of lung cancer.

We firstly determined the activity of Avns on the viability of lung cancer cells. We treated A549 and H1299 cells with different concentrations of AvnA and AvnC (10, 50, 100 μM) in presence or in absence of EGF (25 ng/mL) and we performed the MTT assay after 72 h of exposure. In all of the assays, the known anti-EGFR drug gefitinib (10 µM) was used for comparison.

As shown in Figure 1, AvnC was able to reduce cell viability of both A549 (A) and H1299 (B) cells. Interestingly, when lung cancer cells were treated with EGF, we observed an increased proliferation that was reduced in a dose-dependent manner by both AvnA and AvnC (Figure 1), indicating a possible inhibitory role of Avns in EGF signalling pathway.

To confirm the antiproliferative role of Avns, we performed a clonogenic assay. Cells were incubated with AvnA and AvnC, at different concentrations (10 and 100 µM) with and without EGF (25 ng/mL) for 10 days. Experimental outcomes showed that AvnC was able to significantly restrain the colony formation of cancer cells with results comparable to gefitinib, indicating that this compound is indeed effective in inhibiting proliferation and colony formation ability of lung cancer cells (Figure 2). Interestingly, both Avns were able to inhibit the EGF-induced colony formation, sustaining a possible activity on EGF driven pathway (Figure 1A, Figure 2A,B and Appendix A).

### 2.2. Avenanthramides Inhibit EGFR Induced Lung Cancer Cell Migration

It is well known that normal cells die through apoptosis when detached from extracellular matrix (anoikis) [21]. However, cancer cells undergo phenotypic changes, acquiring the ability to survive and grow under anchorage-independent conditions, as well as to leave the original tumor site, migrate through surrounding tissues and establish metastasis to a distant site. To further investigate the putative anticancer properties of Avns, we analyzed the effects of these compounds on anchorage-independent growth. A549 cells were maintained for 24 or 48 h in suspension and then dead cells were counted. 

Figure 3A,B show that treatment with 100 µM AvnA or AvnC caused an increase in the number of dead cells, both in basal conditions and in EGF-treated cells, indicating that Avns reduce the capability of lung cancer cells to survive in anchorage-independent conditions and inhibit the EGF activity.

In addition, we evaluated the role of Avns on cell mobility and metastatization by studying cell migration and EMT. As shown in Figure 4 and in Appendix A, both AvnA and AvnC did not affect the migration in basal conditions but were effective in reducing A549 cell migration induced by EGF, suggesting that these compounds possess an anti-migratory activity on lung cancer cells.

Given the important role of E-cadherin down-regulation and vimentin up-regulation in phenotypic changes associated to EMT [22,23], we analyzed the expression levels of these proteins on A549 cells after 48 h treatment with Avns in presence/absence of EGF (25 ng/mL). Western blot analysis showed an up-regulation of E-cadherin and a down-regulation of vimentin protein levels upon cell treatment with AvnC (Figure 5 and Appendix A). As expected, EGF was able to both reduce E-cadherin levels and increase vimentin expression, but these effects were counterbalanced by AvnC. Notably, while AvnA increased E-cadherin levels after EGF treatment, we did not observe a similar reduction in vimentin expression (Figure 5). Overall, our data demonstrate that AvnA and AvnC inhibit anoikis, migration, and EMT induced by EGF (although to a different extent). 

### 2.3. EGF-Induced Intrinsic Inflammation Is Inhibited by Avenanthramides

Many works report that intrinsic inflammation promotes cancer development, and that up-regulation of cyclooxygenase 2 (COX-2) plays an important role in tumorigenesis [24,25,26]. Here we evaluated the activity of Avns on COX-2 expression after EGF exposure. A549 cells were treated for 48 h with Avns (10 and 100 µM) with and without EGF (25 ng/mL) and the expression levels of COX-2 were analyzed by western blot. As reported in Figure 6A (and Appendix A), Avns inhibited the COX-2 up-regulation induced by EGF. These data clearly indicate that Avns may inhibit EGF signaling also through the inhibition of COX-2 up-regulation promoted by EGF.

### 2.4. Avenanthramides Affect EGFR Phosphorylation and Signalling 

PI3K/mTOR/Akt and MEK/ERK are two essential EGFR downstream signaling pathways that play key roles in the transduction of proliferative signals from membrane bound receptors [27] and consist of kinases cascades that are regulated by phosphorylation and de-phosphorylation by specific kinases/phosphatases [28].

To verify whether Avns were able to interfere with specific components of EGF pathways we measured the phosphorylation levels of ERK 1/2 and Akt by western blot, using A549 cells pre-treated for 24 h with AvnA and AvnC (10 and 100 µM) and incubated for 15 min with EGF (25 ng/mL). The results showed in Figure 7 (and Appendix A) indicate that both Avns were able to inhibit Akt and ERK 1/2 phosphorylation induced by EGF. However, AvnA was less effective than AvnC.

On the basis of the above reported results indicating that Avns affect EGF effects, we hypothesized that Avns could directly interact with EGFR thus modifying final cell responses.

To support this hypothesis, we verified the ability of Avns to interfere with EGFR phosphorylation induced by EGF. A549 cells were treated with Avns and analyzed by western blotting. The results reported in Figure 8 (and Appendix A) show that both AvnA and AvnC were able to significantly reduce the phosphorylation of EGFR induced by EGF, being AvnC more potent that AvnA, as partially suggested by previous data. These data indicate that Avns are able to reduce lung cancer cells proliferation and migration through the inhibition of EGF-induced EGFR phosphorylation and its signaling pathway. 

To confirm the results produced in lung cancer cells, we repeated some experiments on A431 cell line (squamous cell carcinoma, SCC, cell line), as a model of high EGFR expressing cell line [29]. First, we performed the MTT assay and we observed that also in this model, AvnA and AvnC (10, 50 and 100 µM) reduced both EGF-promoted cell growth (Figure 9A), and EGF-induced EGFR phosphorylation (Figure 9B and Appendix A). As showed in Figure 9C, AvnC is able to inhibit EGFR phosphorylation in a concentration-dependent manner and it appears more active than AvnA with the first active concentration at 0.1 µM (Figure 9C and Appendix A). 

In conclusion, all of these data indicate that Avns are able to inhibit EGFR phosphorylation and EGF signaling pathway. In particular, Avns inhibit tumor growth, migration, EMT and anoikis induced by EGFR activation. AvnC showed to be more active than AvnA in most of the experiments performed.

### 2.5. In Silico Analysis of EGFR/Avenanthramides Interaction

#### 2.5.1. Docking Simulation

Docking simulation and interaction network results revealed that AvnA and AvnC bound with high affinity on EGFR, sharing a similar binding pose in a sensing allosteric binding pocket of the target, named as tyrosine kinase inhibitor (TKI) binding pocket [30], located between the αC helix and the active site, just above the DFG motif, a target region known to accommodate EGFR inhibitors [31] (Figure 10). 

The best-docked conformation of AvnA and AvnC showed Gibbs free-energy values (ΔG) of −7.5 kcal/mol for both ligands. 

Interaction network analyses showed that AvnA formed hydrophobic interactions with Leu-718, Val-726, Ala-743, Lys-745, Leu-788, and Thr-790, four hydrogen bonds with Leu-788, Thr-790, Thr-854 and Asp-855 and a salt bridge with Lys-745 (Figure 10A). On the other hand AvnC trigged hydrophobic interactions with Leu-718, Val-726, Ala-743, Lys-745, Leu-788, and Thr-790, six hydrogen bonds with Leu-788, Thr-790, Cys-797, Thr-854 and Asp-855 and a salt bridge with Lys-745 (Figure 10B). 

To further validate the starting docking pose of our ligands within target binding pocket, we carried out a re-docking simulation for gefitinib on EGFR to compare both the binding affinity and interaction network. 

Docking results and interaction network analyses showed that gefitinib exhibited a Gibbs free-energy value (ΔG) of −7.2 kcal/mol and the ability to form hydrophobic interactions with Lys-745, Thr-790, Leu-792 and Leu-844, a hydrogen bond with Met-793, a water bridge with Thr-854, a halogen bond with Leu-788 and a salt bridge with Asp-800 (Appendix A). 

#### 2.5.2. cMD Simulation

To confirm the docking evidence and to further investigate the stability and potential mechanism of action of Avns on the target, we performed a cMD simulation of 100 ns for EGFR in complex with the two compounds and gefitinib as control. To exclude the presence of potential computational artefacts and showing the reliability of cMD simulation protocols, we computed the backbone structural integrity during the simulations. The target RMSD profiles in bound state showed a stable trend (from 2 Å to 4 Å) along the entire cMD run (Appendix A). To evaluate the stability of the binding pose of compounds within EGFR binding pocket, the RMSD of ligands was computed. The compounds showed a stable trend between 0.5 Å and 3.5 Å (Appendix A). Using the function gmx hbond with the flags –num, -hbn, and -hbm implemented in GROMACS 2019.3, we evaluated the number, the index and the lifetime of hydrogen bonds, as a function of time, of Avns and gefitinib within EGFR binding pocket. The gmx hbond analyses, showed that both the Avns shared a hydrogen bond (occupancy around 100%) between the hydroxyl group belonging to hydroxybenzoic acid group and the Glu-762 side chain. Remarkably, the Anvs also formed a water bridge between the Thr-854 side chain hydroxyl group, the 10666 and 18025 (for AvnA) and 10163 (for AvnC) water molecules and the carbonyl of AvnA and AvnC 1-hydroxyprop-2-en-1-ylidene-amino group, exhibiting an occupancy around 55% and 100%, respectively (Appendix A). Surprisingly, gefitinib showed the same water bridge with Thr-854 (analyzing the co-crystal with 4WKQ PDB code) but with a considerably lower occupancy than Avns within the EGFR binding pocket. Root mean square fluctuation (RMSF) analyses was performed for each biological system to evaluate the influence of compounds, within their binding region, especially, on the αC helix region and the DFG motif. RMSF results showed a superimposable trend for each compound on EGFR binding region (Appendix A). Lastly, to evaluate all possible target-ligand interactions obtained followed cMD and to define the occupancy of each interaction, we performed an analysis of the EGFR/ligand complex trajectory along the entire MD run. Differently by gmx hbond analyses implemented in GROMACS package, where we evaluated potential water bridges formed between water molecule-ligand-target residues, other interaction network analyses were performed using Prolif tool. The Prolif results confirmed the docking results and supported the reliability of the docking starting binding pose. In addition, other remarkable insights were revealed by MD interaction analyses performed with Prolif tool (Figure 11). Interestingly, Avns and gefitinib shared hydrophobic interactions with Leu-718, Val-726, Ala-743, Lys-745, Met-766, Thr-790, Gly-796, Leu-844, Thr-854, and a hydrogen bond with Glu-762 with an occupancy of around 100% of the MD run (Figure 11A–C). Remarkably, only AvnC was able to form a hydrophobic interaction with a very high occupancy against Cys-797 (and a hydrogen bond with a moderate occupancy) and Asp-855 (Figure 11B).

#### 2.5.3. EGFR/Ligand Interaction Energy and Unbinding Pathway (SMD)

To quantify the strength of the interaction of compounds on EGFR, we computed the interaction energy between the protein and the compounds. The total interaction energy (the sum of the Short-Range Lennard-Jones (LJ-SR) and Coulomb (Coul-SR) interaction energy) for AvnA, AvnC, and gefitinib was −198.6 ± 15.3 kJ/mol (−49.65 ± 3.8 kcal/mol), −205 ± 5 kJ/mol (−51.5 ± 1.25 kcal/mol), and −184 ± 7 kJ/mol (−46 ± 1.75 kcal/mol), respectively (Appendix A). Taken together these data suggest that Avns bind spontaneously to the target with high affinity.

To dissect the recognition pathway of compounds to the EGFR, we performed Steered Molecular Dynamics (SMD) simulations. We ran a 500 ps SMD simulation on EGFR in complex with Avns and gefitinib. Both Avns had a steady increase of the applied forces on the first ~150 kJ/mol and ~230 kJ/mol of the simulation, respectively, for AvnC and AvnA, until they reached the maximum, which corresponds to the rupture force of AvnC (300 kJ/mol) and AvnA (400 kJ/mol) unbinding along this dissociation pathway. The force then quickly decreases and stays constant until the end of the simulation. In the first step, between 0 ps and 220 ps of the simulation for AvnC and 0 and 300 ps for AvnA, the two ligands slowly separate and move away from the binding pocket. In the second step, between 220 ps and 30 ps of the simulation for AvnC and 300 ps and 310 ps for AvnA, they move away from the protein and enter the solvent region. Interestingly, gefitinib showed a lower rupture force (200 kJ/mol) than Avns, and it was the first to leave the EGFR binding pocket (250 ps), suggesting a higher inhibitory activity of Avns than gefitinib (Figure 12).

## 3. Discussion

Accumulating evidence suggest that Avns possess antitumoral potential [1] and for AvnA and AvnC specific activity against colorectal cancer [8,15], breast cancer [32,33], and non-small cell lung cancer (NSCLC) [13] has been described.

Lung cancer is a very aggressive tumor associated to a poor prognosis. The curative effects of surgery, radiotherapy, and chemotherapy are insufficient due to individual susceptibly to drugs drug and radiotherapy resistance. Although progress has been achieved in the treatment, the overall survival rate of lung cancer patients is very low and current therapies based on EGFR inhibition exerted by specific tyrosine kinase inhibitors as gefitinib and erlotinib (I generation) or afatinib and dacomitinib (II generation) are not decisive due to the appearance of new EGFR mutations and to side effects. However, EGFR targeting keep a very important role in lung cancer therapy and several efforts have been conducted in order to maximize efficacy and minimize toxicity of drugs [19,20]. 

At present, one fundamental challenge of the research on cancer is the possibility to use natural compounds that often are free from side effects. There is evidence that a number of phytochemicals, including phenolic nutraceuticals, can reduce the metastatic potential of cancer cells by inhibiting EMT pathways. In this context, our efforts aimed to investigate the role of Avns, an important class of molecules derived from oat, as anti-cancer compounds in EGFR-dependent lung cancer progression. In different lung cell models, we demonstrated anti-proliferative, anti-migratory, and anti-EMT activities of AvnA and AvnC, suggesting enhanced functional properties as bioactive nutraceuticals. Importantly, our in vitro studies suggested that AvnA and AvnC inhibit EGFR phosphorylation and signalling. These data were also confirmed on an EGFR overexpressing cell line, demonstrating a possible role of Avns as anticancer compounds active on EGFR pathway.

To further investigate the interplay between Avns and EGFR, computational analysis was performed. Docking simulations showed that Avns bound with high affinity in a sensing region of the target, forming strong interactions with key residues of EGFR. Molecular dynamics simulations confirmed the docking study and showed the high stability and interaction energy of the docked poses. The RMSD of the EGFR/Avns complexes evaluated for the protein backbone and binding pose showed stable profiles and were comparable to gefitinib, demonstrating the ability of Avns to block the target. Molecular dynamics simulation interaction analyses showed that Avns formed a broad hydrophobic and polar network with EGFR binding pocket residues. Interestingly, AvnA and AvnC shared a hydrogen bond with Glu-762 (occupancy around 100%), a highly conserved αC-helix-forming residue that plays a crucial role as a mediating element in EGFR as well as in other kinases [34]. In addition, Thr-854, bound to the DGF motif Asp-855, which is critical for inhibiting EGFR [35,36], was also involved in binding with AvnA and AvnC via a water bridge with lifetimes of 50 ns and 100 ns, respectively. Remarkably, analyzing the interaction network of the EGFR/gefitinib co-crystal (PDB code 4WKQ), we noted that gefitinib triggered the same water bridge with Thr-854 suggesting the critical importance of this binding.

In this work, we have also shown that the inhibitory effect of AvnC is higher than that of AvnA and comparable to that of gefitinib on EGFR. Despite the high chemical-physical and structural feature similarity of Avns, we found differences in the interaction network between the two Avns. First, AvnC showed a higher h-bonds network with EGFR binding residues during the cMD run than AvnA. Second, the water bridge formed by AvnA with Thr-854 is trigged only half of the time compared to AvnC. Such evidence could be explained by the hydrophobic interaction (with a very high occupancy) formed only between AvnC with Cys-797 and Asp-855, which would stabilize the AvnC binding pose, allowing to form a long and stable water bridge, increasing the inhibitory activity of AvnC compared to AvnA. Such evidence would suggest that Cys-797 may represent a novel consensus binding residue for EGFR inhibition. Instead, the similar inhibitory effect of AvnC compared to gefitinib could be explained both by the different chemical-physical properties and different interaction network. In fact, gefitinib is able to form a broader contact network within the EGFR binding pocket than Avns. Furthermore, the RMSF results indicated that Avns equally affected the flexibility of the αC-helix and the DGF motif compared to gefitinib, confirming further in silico findings. However, dissimilarly to AvnA, the RMSF profile of AvnC showed a trend superimposable to gefitinib, suggesting a similar activity on the target, in accordance with in vitro results.

In addition, SMD simulations were performed to investigate the dynamic unbinding processes of Avns from EGFR protein, using as starting structures the last frame obtained following the cMD simulations. Interestingly, AvnC was the compound that left last the target binding pocket, confirming the in vitro results. 

Overall, our results provide an essential molecular basis for the inhibition of EGFR and propose a potential mechanism of action of novel natural inhibitors of EGFR.

Further studies will be necessary to develop a better understand the Avn/EGFR interaction, to evaluate the importance of Avn activity in presence of EGFR mutations, as T790M mutation, and to consider the possibility of applying our observations for pharmacological uses.

## 4. Materials and Methods

### 4.1. Cell Culture

Human NSCLC A549 cancer cell line and epidermoid carcinoma A431 cell line were obtained from ATCC and cultured at 37 °C and 5% CO_2_ in Dulbecco’s modified Eagle’s medium (DMEM), with 4500 mg glucose/l and 100 U/mL penicillin/streptomycin (Euroclone, Milan, Italy), supplemented with 10% fetal bovine serum (FBS, Euroclone, Milan, Italy). Human NSCLC H1299 cancer cell line was cultured in RPMI-1640 (Euroclone, Milan, Italy) following the same conditions.

### 4.2. MTT Assay

A549, H1299 and A431 were plated (2.5 × 10^3^ cells/well) in 96-well multiplates in medium with 10% FBS. After 24 h, cells were starved in 0.1% FBS and then treated with AvnA or AvnC (10, 50, 100 μM) (Sigma-Aldrich, St. Louis, MO, USA) with and without EGF (25 ng/mL) (Peprotech, London, UK) for 48 h in 0.1% FBS. Gefitinib (10 µM) was used as control. Cells were then incubated for 4 h with fresh medium in the presence of 1.2 mM MTT (3-(4,5-dimethylthiazol-2-yl)-2,5-diphenyltetrazolium bromide) (Sigma-Aldrich, St. Louis, MO, USA). The MTT solution was removed and 50 µL of DMSO were added to each well to dissolve the blue formazan crystals. The absorbance of the formazan dye was measured at 570 nm with a microplate reader (EnVision, PerkinElmer, Waltham, MA, USA). Data were expressed as a percentage of the basal control.

### 4.3. Clonogenic Assay 

A549, H1299 and A431 were plated (5 × 10^2^ cells/well) in 6-well multiplates in medium with 10% FBS. After 24 h cells were starved and then treated with AvnA or AvnC (10 and 100 μM) with and without EGF (25 ng/mL) for 10 days in 0.1% FBS. Colonies (>50 cells) were fixed and stained with Diff-Quik, counted and photographed. Data were expressed as a percentage of the basal control.

### 4.4. Western Blotting 

A549 and A431 were plated (2.5 × 10^5^ cells/well) in 6-well multiplates in medium with 10% FBS. After 24 h, cells were treated with AvnA or AvnC (10 and 100 μM) with and without EGF (25 ng/mL) for 48 h or for 15 min in 0.1% FBS.

Cells were then lysated and centrifuged at 15,000× *g* for 15 min at 4 °C. Protein concentration in cell extracts was determined spectrophotometrically using the BCA protein assay kit (Euroclone). Cell extract supernatants containing an equal amount of proteins (50 μg) were treated with Laemmli buffer, boiled for 10 min, resolved on 4–20% stain-free gel and then blotted onto a nitrocellulose membrane using Novablot Semidry System (GE Healthcare Bio-Sciences). The blots were blocked with 5% nonfat dry milk in Tris-buffered saline (TBS) containing 0.5% Tween 20 for 1 h at room temperature and incubated with appropriate dilutions of primary antibodies overnight at 4 °C and subsequently with horseradish peroxidase (HRP)-conjugated secondary antibodies for 1 hr at room temperature. Proteins were then visualized by an enhanced chemiluminescence detection system (EMD Millipore, Burlington, MA, USA). The following antibodies were used: anti-β-actin (Invitrogen, #MA5-11869, LOT XA3487201, 1:500), anti-COX-2 (Cell signaling Technology, Danvers, MA, USA, #12282S, LOT 5, 1:1000), anti-E-cadherin (Cell Signaling Technology, #3195S, LOT 8, 1:1000), anti-GAPDH (Cell Signaling Technology, #5174S, LOT 8, 1:1000), anti-p-Akt (Santa Cruz, SC-514032, LOT H3021), anti-p-Tyr (Cell signaling Technology, #9411S, LOT 27, 1:1000), anti-p-ERK1/2 (Cell signaling Technology, #4370s, LOT 24, 1:1000) or anti-vimentin (Cell Signaling Technology, #5741T, LOT 8, 1:1000). Primary antibodies were detected using affinity-purified HRP-conjugated secondary antibodies (Sigma-Aldrich). Protein bands from western blots were quantified by densitometry using the ImageJ software, and their relative amounts were normalized to the levels of housekeeping proteins serving as internal loading controls.

### 4.5. Scratch Assay

A549 cells were seeded (8.0 × 10^5^ cells/well) in 24-well multiplates in DMEM 10% FBS. After adhesion, cells were starved and after 24 h cells were treated with AvnA or AvnC (100 μM) with and without EGF (25 ng/mL) in DMEM 0.1% FBS. 

Before the treatment with Avns cell monolayers were scored vertically down the center of each well with a sterile tip. Each well was washed with PBS to remove detached cells. Fresh medium (1% serum) with ARA C (Sigma–Aldrich, St. Louis, MO, USA) (2.5 µg/mL) was added to inhibit cell proliferation. Results were expressed as arbitrary units of wound and percentage of healing taking as reference the area at time 0.

### 4.6. Anoikis

Suspension of A549 cells at a density of 5 × 10^5^ cells/mL were incubated in medium with 0.1% FBS (*v*/*v*) and treated with AvnA or AvnC (100 μM) with and without EGF (25 ng/mL) for 24 or 48 h. After incubation, the number of dead cells stained with trypan blue and total cells was evaluated by optical microscope. The number of dead cells was reported as a percentage of total cells.

### 4.7. Structural Optimization and Resources 

3D structure and FASTA sequence of EGFR tyrosine kinase domain 1.85 angstrom structure of EGFR kinase domain with gefitinib (PDB ID 4WKQ, UniProtKB-P00533) was retrieved from the RCSB Protein Data Bank [37] and UniProtKB Database [38], respectively. To avoid errors during the Molecular Dynamic (MD) simulations, missing side chains and steric clashes in PDB files were added/adjusted through molecular modelling, using PyMOD 3.0 and MODELLER v.9.3 [39]. The 3D structure was validated using PROCHECK [40]. GROMACS 2019.3 [41] with charmm36-mar2019 force field [42] was used to resolve high energy intramolecular interaction before docking simulations, and CGenFF [43] was used to assign all parameters to ligands. Structures were immersed in a cubic box filled with TIP3P water molecules and counter ions to balance the net charge of the system. Simulations were run applying periodic boundary conditions. The energy of the system was minimized with 5.000 steps of minimization with the steepest descent algorithm and found to converge to a minimum energy with forces less than 100 kJ/mol/nm. A short 10 ns classic Molecular Dynamics (cMD) was performed to relax the system. All of the cMD simulations were performed integrating each time step of 2 fs; a V-rescale thermostat maintained the temperature at 300 K and Berendsen barostat maintained the system pressure at 1 atm, with a low dumping of 1 ps−1; the LINCS algorithm constrained the bond lengths involving hydrogen atoms.

### 4.8. Docking Simulation 

The last frame obtained by Cmd as described above, was used as reference structure for the docking studies (removing gefitinib from the file). Using Autodock Tools [44], we created a box with dimensions of 25 × 35 × 20 Å, in x, y and z dimensions, able to enclose all gefitinib binding pocket residues. The 3D structures of AvnA and C (CID: 5281157 and 11723200, respectively) were downloaded from PubChem database [45], while the 3D structure of gefitinib was extracted from the PDB file 4WKQ. MGLTOOLS scripts [44] and OpenBabel [46] were used to convert respectively protein and ligand files in pdbqt format, adding gasteiger partial charges. Subsequently, a rational docking for AvnA and C on 4WKQ was carried out, while a re-docking approach was applied for gefitinib on 4WKQ, creating a box able to enclose the docked pose of gefitinib. The docking simulation was performed with Autodock/VinaXB [47], all parameters were used by default. The interaction network of complexes was evaluated by P.L.I.P. Tool [48].

### 4.9. Classical and Steered Molecular Dynamics (SMD) Simulations

All complexes generated by docking simulation were subjected to cMD run of 100 ns, as described above. Prolif tool [49] evaluated all possible target/ligand interactions achieved followed cMD, using all parameters by default.

The pulling simulations necessary to calculate binding energy between the EGFR/Avns and gefitinib complexes were carried out using a 500 ps SMD simulation by Constant Force Pulling of 1000 kJ/mol/nm. The backbone of receptor was kept still, while the ligands were experienced a constant harmonic potential in x, y, z direction, specifically (N, N, Y) for each compound. All ligands were pulled with an external force in the NPT ensemble at 1 atm and 310 K with 2 fs time steps, with a run of 500 ps for each simulation. MD analyses were performed with GROMACS 2019.3 package and displayed with GRACE. PyMOL2.5 was used to generate the 3D structure video and pictures *(**The PyMOL Molecular Graphics System, Version 1.2r3pre, Schrödinger, LLC).*

### 4.10. Protein–Ligand Interaction Energy

To quantify the interaction energy between the target and the compounds we computed the nonbonded interaction energy. GROMACS package allows to evaluate the short-range nonbonded energies using energy grps keyword in the mdp file. The energy terms of interest are the average short-range Coulombic interaction energy (Coul-SR) and the short-range Lennard–Jones energy (LJ-SR). The total interaction energy (IE_Binding_) is defined by the equation:IE_Binding_ = Coul − SR + LJ − SR

### 4.11. Statistical Analysis

Data were generated from independent experiments as reported in figure legends and expressed as means ± standard deviation (SD). Statistical analysis was performed using Student’s t test for unpaired data (Graph Pad-Prism); *p* < 0.05 was considered statistically significant.

## Figures and Tables

**Figure 1 ijms-23-08534-f001:**
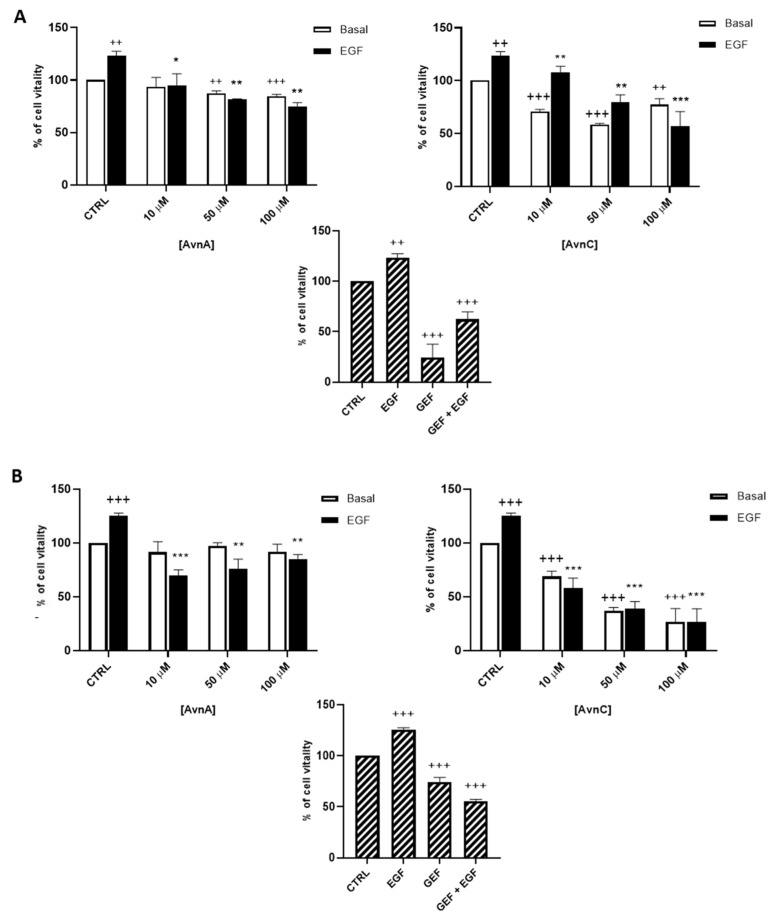
Effect of natural Avns on lung cancer cell vitality. A549 (**A**) and H1299 (**B**) vitality was evaluated by the MTT assay. Cells were exposed to EGF in presence of increasing concentrations of Avns (10, 50, and 100 µM) for three days. Data are expressed as % over basal control and are representative of three independent experiments run in triplicate. Statistical analysis: +++ *p* < 0.001 and ++ *p* < 0.01 vs. CTRL; *** *p* < 0.001; ** *p* < 0.01 and * *p* < 0.05 vs. EGF.

**Figure 2 ijms-23-08534-f002:**
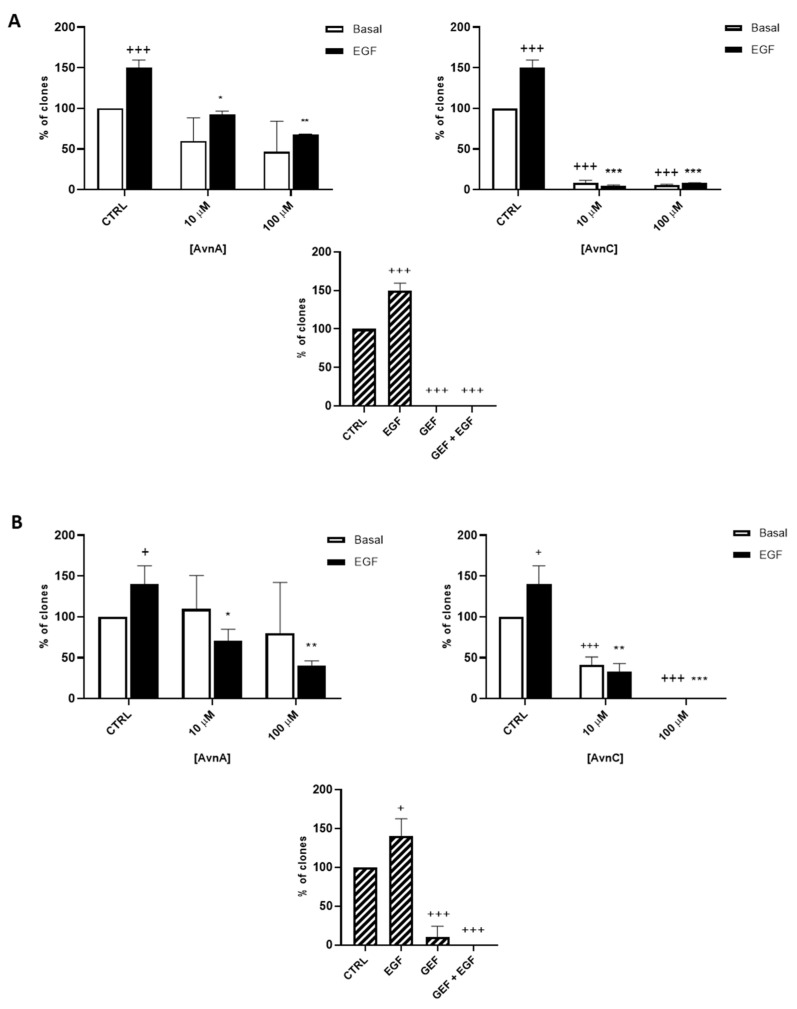
Natural Avns reduce clonogenicity of lung cancer cells promoted by EGF. Percentage of colonies of A 549 (**A**) and H1299 (**B**) cells in response to EGF in presence or absence of Avns. Data are expressed as % over basal control and are representative of three independent experiments run in triplicate. Statistical analysis: +++ *p* < 0.001 and + *p* < 0.05 vs. CTRL; *** *p*< 0.001; ** *p*< 0.01 and * *p* < 0.05 vs. EGF.

**Figure 3 ijms-23-08534-f003:**
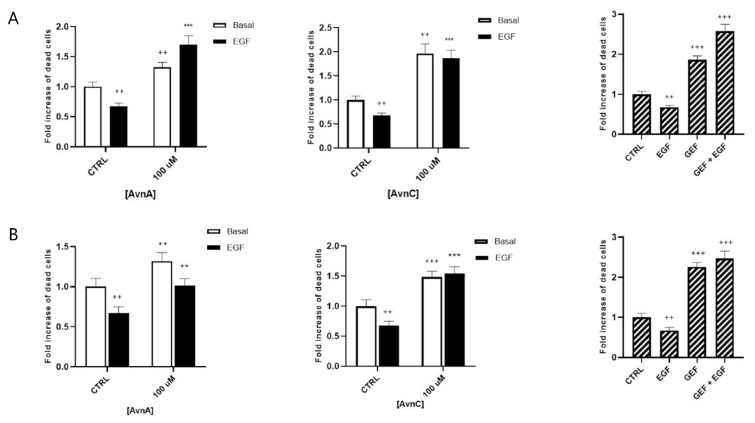
Effect of Avns on anoikis after 24 (**A**) and 48 (**B**) h. Cell vitality of A549 in suspension treated with Avns (100 µM) in 0.1% of serum. Results are expressed as fold of increase of dead cells and are representative of three independent experiments run in triplicate. Statistical analysis: +++ *p* < 0.001 and ++ *p* < 0.01 vs. CTRL; *** *p* < 0.001 and ** *p* < 0.05 vs. EGF.

**Figure 4 ijms-23-08534-f004:**
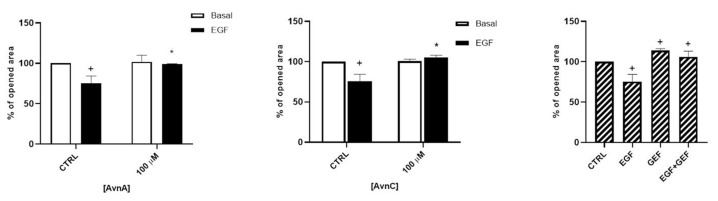
Avns reduce migration of tumor cells promoted by EGF. A549 migration was evaluated by scratch assay. Data are reported as % of opened area over basal control and are representative of three independent experiments run in triplicate. Statistical analysis: + *p* < 0.05 vs. CTRL and * *p* < 0.05 vs. EGF.

**Figure 5 ijms-23-08534-f005:**
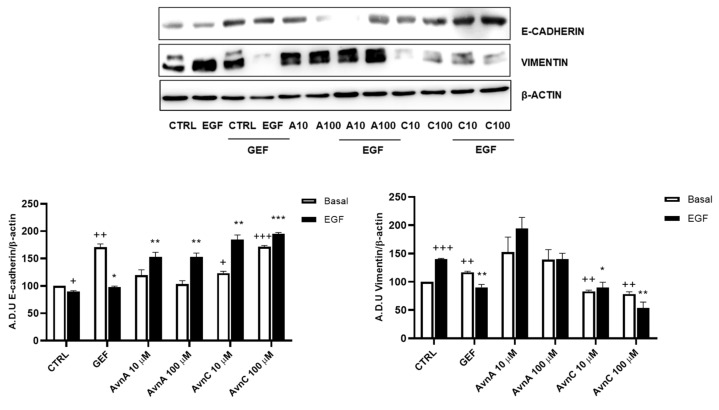
Avns regulate E-cadherin and vimentin expression. Representative images and quantification of western blot analysis of A549 cells exposed to Avns (10 and 100 µM, 48 h) (A.D.U: arbitrary densitometry units). Data are expressed as fold increase compared with control and are representative of three independent experiments. Statistical analysis: +++ *p* < 0.001; ++ *p* < 0.01 and + *p* < 0.05 vs. CTRL; *** *p* < 0.001; ** *p* < 0.01 and * *p* < 0.05 vs. EGF.

**Figure 6 ijms-23-08534-f006:**
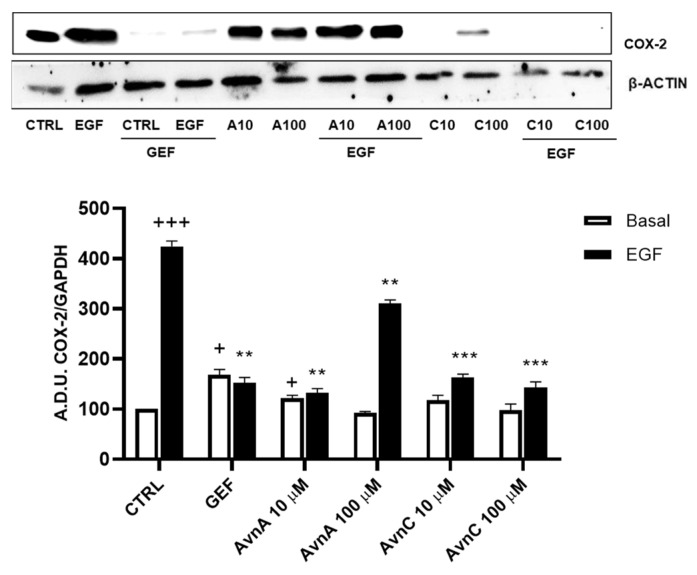
Regulation of inflammation markers by Avns. Representative images and quantification of western blot analysis of COX-2 expression in A549 cells treated with Avns (10 and 100 µM, 48 h). (A.D.U: arbitrary densitometry units). Data are expressed as fold increase compared to control and are representative of three independent experiments. Statistical analysis: +++ *p* < 0.001 and + *p* < 0.05 vs. CTRL; *** *p* < 0.001 and ** *p* < 0.01 vs. EGF.

**Figure 7 ijms-23-08534-f007:**
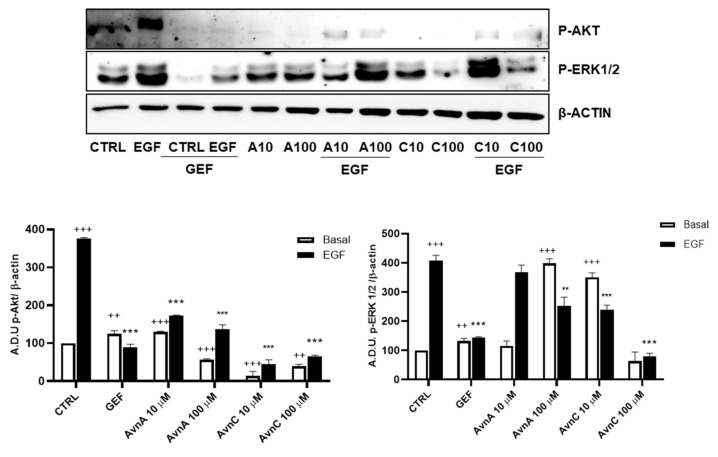
Induction of phosphorylated proteins by Avns. Representative images and quantification of western blot analysis of Akt and ERK 1/2 phosphorylation in A549 cells pre-treated with Avns (10 and 100 µM, 24 h) and 15 min with EGF (25 ng/mL). (A.D.U: arbitrary densitometry units). Data are expressed as fold increase compared to control and are representative of four independent experiments. Statistical analysis: +++ *p* < 0.001 and ++ *p* < 0.01 vs. CTRL; *** *p* < 0.001 and ** *p* < 0.01 vs. EGF.

**Figure 8 ijms-23-08534-f008:**
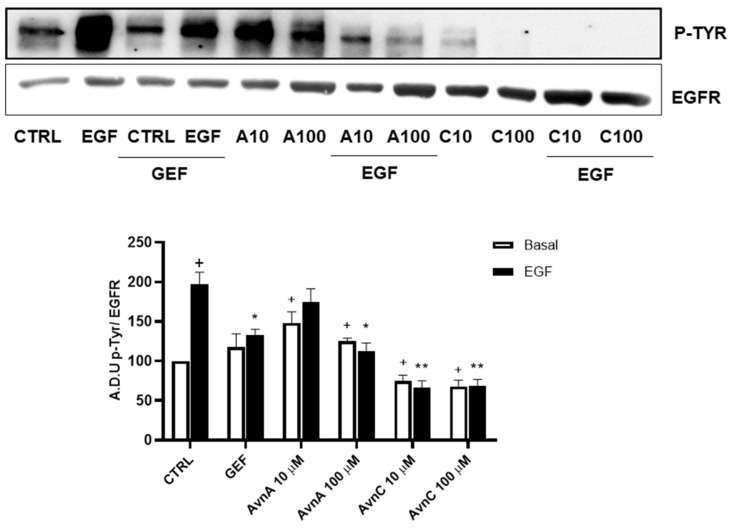
Regulation of EGFR phosphorylation by Avns. Representative images and quantification of western blot analysis of p-EGFR in A549 cells treated with Avns (10 and 100 µM, 48 h). (A.D.U: arbitrary densitometry units). Data are expressed as fold increase compared to control and are representative of three independent experiments. Statistical analysis: + *p* < 0.05 vs. CTRL; ** *p* < 0.01 and * *p* < 0.05 vs. EGF.

**Figure 9 ijms-23-08534-f009:**
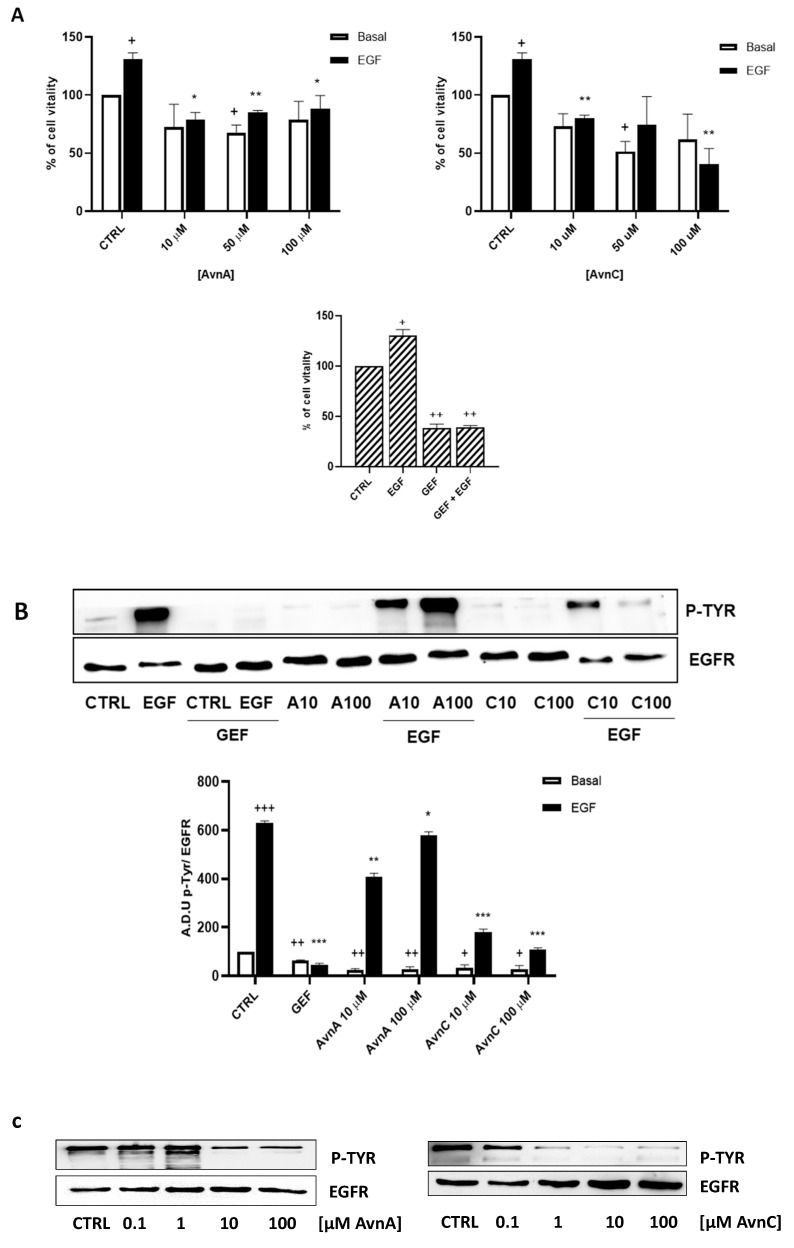
Effects of Avns on A431 cells vitality and EGFR phosphorylation. (**A**) MTT assay. Cells were exposed to EGF in the presence of increasing concentrations of Avns (10, 50, and 100 µM) for three days. Data are expressed as % over basal control and are representative of three independent experiments run in triplicate. Statistical analysis: ++ *p* < 0.01 and + *p* < 0.01 vs. CTRL; ** *p* < 0.01 and * *p* < 0.05 vs. EGF. (**B**) EGFR phosphorylation. Representative images and quantification of western blot analysis in A431 cells treated with EGF in presence or absence of Avns (10 and 100 µM). (A.D.U: arbitrary densitometry units). Data are expressed as fold increase compared to control and are representative of three independent experiments. Statistical analysis: +++ *p* < 0.01; ++ *p* < 0.1 and + *p* < 0.5 vs. CTRL; *** *p* < 0.001; ** *p* < 0.01 and * *p* < 0.05 vs. EGF. (**C**) Representative images and quantification of western blot analysis in A431 cells treated with Avns. Data are representative of three independent experiments.

**Figure 10 ijms-23-08534-f010:**
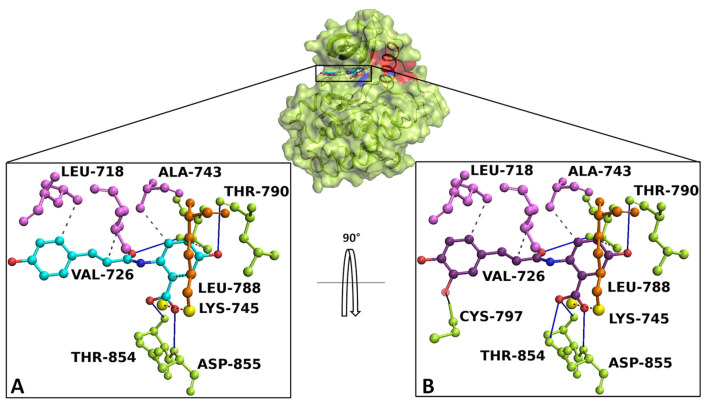
Overview of EGFR docked with AnvA and AvnC. The EGFR 3D structure is depicted in green surface/cartoon. The αC helix region and DFG motif are represented in red and blue surface/cartoon, respectively. AvnA and AvnC were reported as cyan and purple sticks/balls, respectively. The enlargement shows the interaction network of (**A**) AvnA and (**B**) AvnC in complex with the binding residues of EGFR after the docking simulation. The residues involved in hydrogen bonds (blue continue line), salt bridge (red dotted line), and hydrophobic interactions (grey dotted line) are indicated as green, orange, and violet sticks/balls, respectively. The yellow sphere, represent the charge centre of atoms forming the salt bridge. To clarify, the hydrogens were hidden.

**Figure 11 ijms-23-08534-f011:**
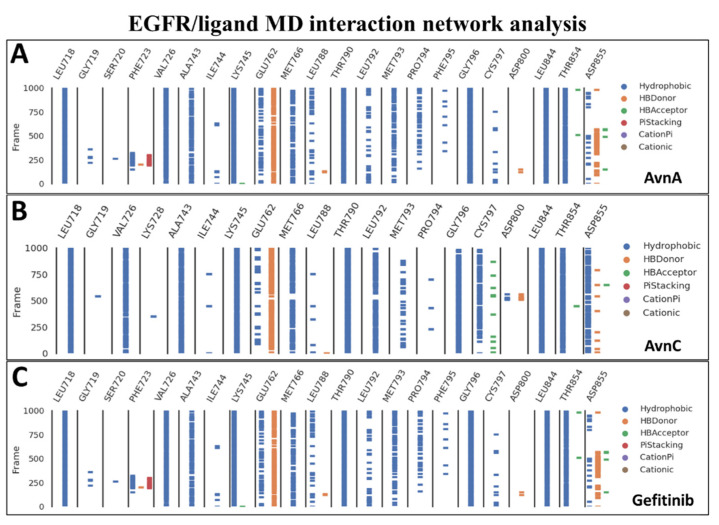
MD interaction network analysis. The binding residues and the molecular dynamics run time (10,000 frames of 100 ns of MD run time) are reported on the *x* and *y* axis, respectively. The hydrophobic, π-stacking, cationic, π-cationic interactions, and the hydrogen bonds (HB-Donor and HB-Acceptor) are reported as blue, red, brown, purple, orange, and green lines, respectively for (**A**) AvnA, (**B**) AvnC, and (**C**) gefitinib.

**Figure 12 ijms-23-08534-f012:**
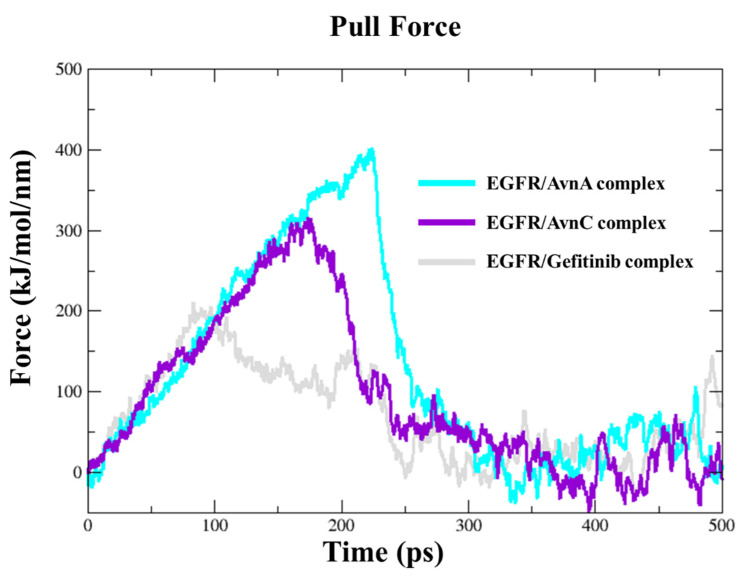
Steered molecular dynamics simulations. Force profiles of compounds pulled out of the EGFR binding pocket along the unbinding pathway, AvnA (cyan line), AvnC (purple line), and gefitinib (grey line).

## Data Availability

Not applicable.

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
