# Peer review of "Pharmacological and In Silico Analysis of Oat Avenanthramides as EGFR Inhibitors: Effects on EGF-Induced Lung Cancer Cell Growth and Migration"

_ijms, 2022, doi:10.3390/ijms23158534_

Round 1

Reviewer 1 Report

Please see attached report

Author Response

# General considerations

First, the reviewer wishes to express their gratitude to the editors for giving them the opportunity to help the authors and to the authors for sharing their research.

The manuscript assesses the ability of two members of a class of naturally occurring compounds from Avena Sativa, AvnA and AvnC as EGFR-targeted tumour-suppressing drugs, using NSCLC cell lines as test beds for their effectiveness against EGFR-directed tumour growth at various levels and then analysing the molecular determinants of their binding through two different methods of molecular dynamics simulation.
Gefitinib, a 1st generation, class I reversible anti-EGFR TKI is used as a benchmark throughout and the authors conclude that the effectiveness of Avns, in particular of AvnC, is broadly comparable to that of gefitinib, and so is their binding affinity as calculated by MDS.

Major Issues

General

The cell line used through most of the paper is A549, a NSCLC cell line that expresses EGFR and where gefitinib is effective in suppressing cell growth and EGFR signalling.
However, it must be noted that A549 is not the ideal model for anti-EGFR therapy in NSCLC, as it expresses WT EGFR (while the classical clinical indication for anti-EGFR TKI is expression of kinase-mutated EGFR) and also expresses a KRAS G12S mutation, which is considered a negative indication for anti-EGFR therapy.
https://web.expasy.org/cellosaurus/CVCL_0023

https://cancer.sanger.ac.uk/cosmic/mutation/overview?id=100251480

Now, it is clear from the results that gefitinib is effective in vitro in this cell line, and that the chosen Avns also are, but it is unclear to me why this cell line was selected instead of one that more closely mimics tumours that would be selected for anti-EGFR targeted therapy.

The first two figures also included NSCLC cell line H1299, which I guess must be NCI-H1299 (https://web.expasy.org/cellosaurus/CVCL_0060), and which is also a WT EGFR NSCLC cell line.

It is unclear from the text why this second cell line was dropped. Perhaps the authors can explain?

This observation of the Reviewer is very relevant since EGFR inhibitors are used in clinical practices when EGFR is mutated. However, this manuscript represents a first tentative to demonstrate the activity of Avns as potential EGFR inhibitors. In our model we used EGF as external stimulus to induce EGFR activation and to demonstrate whether Avns were able to block EGF/EGFR pathway activation. Of course, in the light of the results obtained in this work, the next step of our research will be to evaluate the activity of Avns on EGFR mutated cancer cells. In effect we have some initial observations by computational analysis that AvnC may inhibit EGFR with T790 mutation.

At the beginning of this manuscript, we showed both A549 and H1299 cell lines to demonstrate that Avns inhibit EGF-induced cell vitality and clonogenicity in two different cell lines. However, due to the same characteristics of these cell lines (both expressing WT EGFR) we continued only with A549, because we did not have any additional information deriving from the use of both lines. Instead, we used A431 cell line to verify the activity of Avns on constitutively active EGFR.

Figure 4

The data report on a wound healing assay, however, looking at the data, it appears that EGF inhibits wound closure compared to no treatment, and gefitinib promotes wound healing, which is something that goes counter to what normally happens with these assays.
See for example
https://www.researchgate.net/figure/EGF-stimulates-cell-migration-A-Wound-healing-assay-was-made-in-both-cell-lines-with_fig6_236598346
Can the authors double check if this is what they meant to plot, or if there has been some mistake?
If there were no plotting/normalisation mistakes, can the authors explain what is going on.
Also, can they please add to supplementary materials representative images of the wound healing assay?

We thank the Reviewer for the observation. There has been a mistake in the preparation of the figure. In the revised form of the manuscript the title of Y axis was changed with “% of opened area”. In addition, representative images of the scratch assay have been added to supplemental materials (new figure S2).

Results section #2.5.2

Can the authors please plot the data on the differences in Thr854 bond occupancy between Avns and gefitinib and the significance of the difference? Looking at Figure 11, which also reports on occupancy, the coloured bars for Thr854 look essentially the same between AvnA and gefitinib, while AvnC has one fewer green bar.

We thank the Reviewer for the useful suggestion. Providing in-depth information about the analyses carried out in a study is very important to make the study clearer. In this work, several structural and energetic analyses were performed, to dissect both the potential binding mode and potential mechanism of action of our compounds against the target. The interaction network trigged between an active compound and the target, it could represent important evidence to propose a potential docking pose and mechanism of action of a ligand on the target. Here, different analyses were performed to evaluate the potential activity of compounds against the target, among them, we also focused on the formation of a water bridge between the ligand and Thr-854 residue of the target (because such water bridge is present in the experimental co-crystal of the 1.85 angstrom structure of EGFR kinase domain with gefitinib, PDB CODE 4WKQ). As reported in the manuscript, two different analyses were performed to evaluate the interaction network of the MDs between our compounds and the target. Firstly, using Prolif tool, we evaluated all interactions trigged between the protein residues and our compounds (Figure 11), where no significant information was observed between Avns and Gefitinib with Thr-854, second, using GROMACS package, through the gmx hbond function with -hbn, -hbnum and -hbm flags, we evaluated the lifetime of potential water bridges between water molecule-ligand-target residues. The gmx hbond function implemented in GROMACS provided us interesting results, where Avns were able to form water bridges with Thr-854 showing an occupancy very high (as shown in Video S1 A-B), while no water bridge with high occupancy was detected for gefitinib in complex with the target. The goal of our analyses was not to identify a hydrogen bond between our compounds and Thr-854 (analysed by Prolif tool), but to detect potential water bridges between the water molecule-ligand-target Thr-854 (analysed by gmx hbond with GROMACS). However, we agree with the Reviewer about the poor clarity of the sentence. The authors rewrote the sentence to make clearer the concept and the goal of the analysis.

Materials and Methods #4.6

The description of the method appears to be incomplete. It is not obvious how the method would report on anoikis. Were the cells grown on soft agar? What timings and concentrations were used for the drugs?

We amended the section #4.6 in the new version of the revised manuscript.

Minor Issues

Introduction

Reference #19 is specifically about EMT, however EMT is not the only resistance mechanism to targeted anti-EGFR therapy. I would suggest to change this reference to a review with a broader scope, something like:

  1. Asao, T., Takahashi, F. & Takahashi, K. Resistance to molecularly targeted therapy in non-small-cell lung cancer. Respiratory Investigation (2018) doi:10.1016/j.resinv.2018.09.001.

2.Nagano, T., Tachihara, M. & Nishimura, Y. Mechanism of Resistance to Epidermal Growth Factor Receptor-Tyrosine Kinase Inhibitors and a Potential Treatment Strategy. Cells 7, 212 (2018).

Also, at lines #47-50, you state that Avns inhibit a series of cancer processes, including apoptosis. Wouldn’t it be more correct to say that Avns regulate apoptosis?

We changed reference 19 by adding both the references suggested by the Reviewer. We also specified in the text that Avns “regulate apoptosis”.

All figures

Please indicate in a non-ambiguous way what the number of repeats is. In Fig 1, for example you say the result comes from 3 independent experiments done in triplicate, so is total n = 9?

We thank the Reviewer for this question, and we would like to better explain what we did.

For functional assays (Fig. 1-4)  the sentence “representative of three independent experiments run in triplicate” means that we repeated each experiment 3 times using cells at different passages or using different clones. In each experiment we have tripled each specific experimental point to minimize errors linked to cell counting or related to the procedure, and for each experimental points we reported the average of the triplets. Thus, based on what has just been said, n (i.e. the number of independent experiments) is 3 and not 9.

Figure 2

Could the authors please add to the supplementary material representative images of the plates for the clonogenicity assay?

Representative images of clonogenic assay have been added as Figure S1 of supplemental materials.

Figure 3

The normalisation of the data makes things a bit strange because the negative control is set at 100%, so for the inhibitors it looks as if more cells died than there were originally, which is a bit weird.
Can the authors change their normalisation to maybe fold change instead of percentage to avoid this?

Figure 3 has been modified accordingly to the Reviewer’s suggestion. In the new version of the manuscript data are expressed as fold change.

Figures 5-9

Could the authors add to the Supplementary Material representative raw images of their blots?

Raw images of blots have been added in supplemental materials (Figures S3-S8).

Results section #2.4

At line #230, in the text describing the results of Fig 9B, it appears that a connecting structure between the two halves of the sentence is missing, impairing readability and comprehension.
Can the authors please revise?

We rewrote the sentence to make it more understandable.

Results section #2.5.1

In the first paragraph, the pocket you’re describing is the usual TKI binding pocket, isn’t it? Can you specify it?

Thanks to the Reviewer for the observation. To provide in-depth and detailed definitions about a target structural feature as well as target-ligand interaction is a crucial step to make clearer the manuscript. The authors agree with the Reviewer and rewrote the sentence.

Figure 10
The colours of the dotted lines are hard to see against the rest of the figure.

The resolution of picture is very important to make the paper clearer. As suggested by the Reviewer, we re-organized our picture and provided them in high resolution (Figure 10).

Results section #2.5.3
Can you please plot the data regarding the total interaction energy (written out at lines #335-337) and show the significance thereof?

To provide both detailed information and the theory about the analyses carried in a study is a critical step to make clearer and reproducible the manuscript. The authors agree with the Reviewer about the poor information provided for the interaction energy analyses. As suggested by the Reviewer, we added the data regarding the total interaction energy as a plot (Figure S11) and showed the significance of the analysis.

Discussion

References missing at line #369 on EGFR TKI resistance and at line #371 on efforts to overcome it. You could re-use the references I suggested for the introduction.

We thank the Reviewer for this suggestion. We added both references in the revised version of the manuscript.

Materials and methods Sections #4.2-4.5

The references are unnecessary as the methods are completely described and of common use. Please remove.

All the references have been removed from this section.

Materials and Methods Section #4.4

Please state the product code, clone number and dilution for all antibodies.

The product code, clone number and dilution for all antibodies have been added in the revised version of the manuscript.

Materials and Methods Section #4.7

Do the force field and CGenFE need to have their own citations?

Also, citation #39 appears to be the wrong one and does not point towards RCSB PDB.

We added charmm36-mar2019 and CGenFE force field citations and corrected the RCSB PDB citation.

Materials and Methods Section #4.9

Please cite PyMOL so the project can keep on getting funded.

We added the PyMOL citation.

Materials and Methods Section #4.10

In some cases more than 3 replicates appear to have been sued. Please amend to reflect this.
What software has been used to perform the statistical analysis? Can you please state it?

The section 4.10 of materials and methods have been corrected as suggested by the reviewer.

Data availability statement

In the interest of reproducibility, you might want to say that you’re going to provide the raw data/ the compounds/the simulation files on reasonable request?
You might also want to consider posting the raw WB and images for the various assays on a general repository for raw data such as Zenodo, Figshare or Dryad.

We agree with the Reviewer for this suggestion and of course the raw data will be available after reasonable request. Meanwhile, all the raw data of the representative images showed in the manuscript have been added to the supplemental materials file as requested by reviewer 1 and 2.

Language and Typos

  • Line #27 – nearby to a region in the vicinity of
  • Line #264 – while On the other hand
  • Line #304 – belonged belonging
  • Line #310 – occupancy lower that lower occupancy than
  • Line #318 – in addedition in addition
  • Line #406 – half of the time
  • Line #409 – rising increasing
  • Line #421 – leaved for left
  • Line #427 – to apply of applying
  • Line #517 – above mentioned as described above
  • Line #532 – above mentioned as described above

The authors modified all the language and typos as suggested by the Reviewer.

Reviewer 2 Report

Comment

The manuscript by Trabalzini et al. presented an anti-cancer effect of avenanthramide. They presented some impressive data, but there are some question and suggestions as described below.

1.       The author should mention why the author test the effect of AvnA, AvnC but not AvnB.

2.       The author should provide original picture of these experiments in the manuscript including

2.1. Clonogenic assay in figure 2,

2.2 Scratch assay in figure 4

3.       In some experiment, the authors conducted experiment by using Avns only 1 or 2 doses,

The author should mention the reasons why conducted only these doses.

At least 3 doses should be conduct to see dose- and time-dependent effects.

4.       The author should test the effect of Avns on normal cells in comparison with cancer cells to see cytotoxic effect.

5.       It is very difficult to see western blot result in figure 5,

the author should try to set it in the same position or draw the line to separate each lane.

6.       GAPDH shows not equal band intensity. The authors should change to beta-actin similar to other experiments.

7.       It could be nice if the authors provide graphical abstract for this study, it will help reader to understand more easily

Author Response

Comment

The manuscript by Trabalzini et al. presented an anti-cancer effect of avenanthramide. They presented some impressive data, but there are some question and suggestions as described below.

  1. The author should mention why the author test the effect of AvnA, AvnC but not AvnB.

We would like to thank the Reviewer for this question, and we are going to explain why we have studied in this work only AvnA and AvnC and not AvnB which indeed, together with the previous two, is the most abundant Avn in nature.

In a previous work focused on the study of the biological activity of two novel avenanthramides produced in recombinant yeast and named Yav I and Yav II (Moglia A, et al., Evaluation of the bioactive properties of avenanthramide analogs produced in recombinant yeast. Biofactors. 2015;41(1):15-27. doi: 10.1002/biof.1197) we had actually used AvnB as comparison.  However, in the following work aimed to evaluate the antitumor potential of natural and yeast Avns in colon cancer cells (Finetti F, et al., Yeast-Derived Recombinant Avenanthramides Inhibit Proliferation, Migration and Epithelial Mesenchymal Transition of Colon Cancer Cells.Nutrients. 2018;10(9):1159. doi:10.3390/nu10091159), we moved to AvnA and Avn C due to their high  structural analogy with YavI and YavII respectively. For consistency and continuity with this previous work, we continued by extending the study of the antitumor activity of AvnA and AvnC in lung cancer cells.

It must be also remembered that, as can be easily verified by performing a search in PubMed, AvnC is among all the Avenanthramides the most studied and most active in cancer. Also, in our hands AvnC appears the most active in different cancer cell models, while AvnA, that differs from AvnC in the absence of an hydroxyl group in the 3’ position, is a good compound to compare to AvnC for structure-function relationship studies.

However, based on previous studies and also in the light of what it has been obtained in this work, it will be of fundamental importance to evaluate the activity of AvnB and other natural or semi-synthetic Avns in a future work.

  1. The author should provide original picture of these experiments in the manuscript including
    • Clonogenic assay in figure 2,

2.2 Scratch assay in figure 4

As suggested, we added in the supplemental material section raw data of all the western blotting images reported in the manuscript and an original representative picture of both clonogenic and scratch assay.

  1. In some experiment, the authors conducted experiment by using Avns only 1 or 2 doses,

The author should mention the reasons why conducted only these doses.

At least 3 doses should be conduct to see dose- and time-dependent effects.

We agree with the Reviewer that at least 3 doses are necessary to perform concentration-dependent curves. In fact, the first experiments aimed at determining the effects of Avns on cell viability (Figure 1) were carried out using three different concentrations of AvnA and AvnC. However, since, as it is evident in figure 1, no significant differences were found between the 50 and 100 µM concentration, we decided to continue with only the 1 and 100 µM concentrations or, in some cases, with the highest concentration of the two Avns.

  1. The author should test the effect of Avns on normal cells in comparison with cancer cells to see cytotoxic effect.

This is a very important point. In our previous paper aimed to study the effect of natural and yeast Avns in colon cancer cells (Finetti F et al., Yeast-Derived Recombinant Avenanthramides Inhibit Proliferation, Migration and Epithelial Mesenchymal Transition of Colon Cancer Cells.Nutrients. 2018 Aug 24;10(9):1159. doi:10.3390/nu10091159) we used colon cancer cell lines and CCD18 (human fibroblast cell line isolated from normal colon tissue) for comparison, demonstrating that the activity of Avns was selective towards cancer cells. Based on these already published results, we didn’t show the data on non-cancerous cells again.

  1. It is very difficult to see western blot result in figure 5,

the author should try to set it in the same position or draw the line to separate each lane.

As suggested by the Reviewer, the figure 5 has been amended

  1. GAPDH shows not equal band intensity. The authors should change to beta-actin similar to other experiments.

Figure 6 has been corrected. A new blot of COX-2 with its corresponding β-actin has been inserted in the figure.

  1. It could be nice if the authors provide graphical abstract for this study, it will help reader to understand more easily

A graphical abstract summarizing the study had been uploaded during the first submission of the manuscript. We submit it again together with the revised version of the manuscript to show it to the reviewer.

Reviewer 3 Report

Lorenza Trabalzini et al. discuss in their article the effect Avenanthramide A and C can have on the epithelial growth factor (EGF) induced aggressiveness of lung cell carcinoma cells. Avenanthramide A and C are components of Avena sativa L.:  oat. 

The authors demonstrate that in cell cultures Avenanthramide C has a negative effect on lung cancer cell vitality, their tendency to form colonies and their ability to migrate (cells A549 and H1299). In most cases these effects can not be reversed by adding EGF. 

Correspondingly biochemical effects induced by EGF, like the increase of inflammation markers and the activation of EGF downstream kinase cascades are partially reduced under the effect of Avenanthramide C and A. In particular the phosphorylation of the EGF receptor is remarkably reduced. 

These experimental data are supported by a binding simulation of Avenanthramide A and C to the EGF receptor. The simulation suggests that Avenanthramide A and C bind in the EGF receptor binding pocket and that the force to remove them is higher than for Gefitinib, a commonly used tyrosine kinase inhibitor. 

Taken all together the presented data suggest that Avenanthramide C is a bioactive component which might be exploitable in an anti-cancer therapy. 

The manuscript can be published as it is. 

Author Response

Lorenza Trabalzini et al. discuss in their article the effect Avenanthramide A and C can have on the epithelial growth factor (EGF) induced aggressiveness of lung cell carcinoma cells. Avenanthramide A and C are components of Avena sativa L.:  oat. 

The authors demonstrate that in cell cultures Avenanthramide C has a negative effect on lung cancer cell vitality, their tendency to form colonies and their ability to migrate (cells A549 and H1299). In most cases these effects can not be reversed by adding EGF. 

Correspondingly biochemical effects induced by EGF, like the increase of inflammation markers and the activation of EGF downstream kinase cascades are partially reduced under the effect of Avenanthramide C and A. In particular the phosphorylation of the EGF receptor is remarkably reduced. 

These experimental data are supported by a binding simulation of Avenanthramide A and C to the EGF receptor. The simulation suggests that Avenanthramide A and C bind in the EGF receptor binding pocket and that the force to remove them is higher than for Gefitinib, a commonly used tyrosine kinase inhibitor. 

Taken all together the presented data suggest that Avenanthramide C is a bioactive component which might be exploitable in an anti-cancer therapy. 

The manuscript can be published as it is. 

The authors are very grateful to the Reviewer for the positive evaluation of their manuscript.

Round 2

Reviewer 2 Report

The manuscript has been sufficiently improved and is now ready to be published.